# Protein Hydration in a Bioprotecting Mixture

**DOI:** 10.3390/life11100995

**Published:** 2021-09-22

**Authors:** Silvia Corezzi, Brenda Bracco, Paola Sassi, Marco Paolantoni, Lucia Comez

**Affiliations:** 1Dipartimento di Fisica e Geologia, Università degli Studi di Perugia, 06123 Perugia, Italy; silvia.corezzi@unipg.it; 2Dipartimento di Chimica, Biologia e Biotecnologie, Università degli Studi di Perugia, 06123 Perugia, Italy; brenda.bracco@studenti.unipg.it (B.B.); paola.sassi@unipg.it (P.S.); 3CNR-IOM at Dipartimento di Fisica e Geologia, Università degli Studi di Perugia, 06123 Perugia, Italy

**Keywords:** hydration water, light scattering, lysozyme, trehalose

## Abstract

We combined broad-band depolarized light scattering and infrared spectroscopies to study the properties of hydration water in a lysozyme-trehalose aqueous solution, where trehalose is present above the concentration threshold (30% in weight) relevant for biopreservation. The joint use of the two different techniques, which were sensitive to inter-and intra-molecular degrees of freedom, shed new light on the molecular mechanism underlying the interaction between the three species in the mixture. Thanks to the comparison with the binary solution cases, we were able to show that, under the investigated conditions, the protein, through preferential hydration, remains strongly hydrated even in the ternary mixture. This supported the water entrapment scenario, for which a certain amount of water between protein and sugar protects the biomolecule from damage caused by external agents.

## 1. Introduction

Biopreservation is the answer for protecting at-risk biological material. Such a process aims to maintain very low the biological activity of a biomolecule during possible long-term stasis, without loss of viability once the temperature and humidity are restored at physiological levels. This can be achieved either by dehydrating through, e.g. freeze-drying procedures, or by reducing temperature (cryo-preservation). The addition of different solutes (or cosolvents) helps to stabilize both proteins and membranes during freezing. Sugars, polyalcohols, several salts, peptides, and even proteins are particularly well-suited for this use. The non-specific effect key to the success of so many different stabilizers may be ascribed to the preferential exclusion mechanism [1,2], i.e., to the evidence that the solutes that stabilize biomolecules are those preferentially excluded from around the protein. This exclusion has an entropy cost related to the extension of the solvent-exposed surface area of the protein. Thus, the stabilization of the folded state is because its surface area is lower than that of the unfolded one.

Among different sugars, trehalose has a special place [3,4,5,6,7,8,9]. It is known to be more effective than other sugars to preserve the functionality of biomolecules under stress conditions that would naturally promote their lability or denaturation [1,10,11]. Nature has also taught us that some plant and animal cells spontaneously produce trehalose to survive under extreme dehydration conditions, allowing dried microorganisms to enter and exit dormancy [12]. Trehalose protects biomolecules in vivo and has the same protective ability in vitro, which has opened new fields of application in food preservation technologies and pharmaceutical manufacturing processes. This discovery has led to a proliferation of studies on the subject. Several molecular mechanisms, often complementary, have been proposed to explain the remarkable effectiveness of trehalose in preserving biomolecules. The hypotheses vary from trehalose that is presumed to substitute water in the protein hydration shell by forming direct hydrogen bonds with the hydrophilic sites of the biomolecule (water replacement [13]), to trehalose that is supposed to retain the protein hydration and native structure by water molecules close to the biomolecular surface (water entrapment [14,15,16,17]). In ternary solutions, these mechanisms, associated with different preferential interactions among the species present in the mixtures [18], may strongly depend on the environmental conditions, such as temperature, pressure, ionic strength, and sugar concentration. Sugar molecules force their hydrogen-bonding imprint on the water network above a specific concentration (30% in weight) [19].

Consequently, the protein will tend to interact preferentially with surrounding water molecules rather than with sugar. This causes the slowing down of hydration water dynamics and local protein motions, which is considered a relevant contributing factor to biopreservation [20]. A recent molecular dynamics study on a water/trehalose/lysozyme mixture (around 40 wt. % trehalose in the global system) also showed that the dynamics of water proceeds on different time-scales and that cooling remarkably enhances the slowdown of water molecules in close vicinity to the protein surface [16]. Such an effect is supposed to inhibit ice formation and enable vitrification without biological damage, thus indicating the cryoprotectant origin of trehalose [21]. Our group contributed to this research by using the Extended frequency range Depolarized Light Scattering (EDLS) technique, which has proved to be a powerful tool to probe the fast dynamics of water, allowing one to separate the solute from solvent dynamics [22,23,24] and bulk from hydration water contributions [22,25,26]. Our results for the same mixture as that investigated in Reference [16] provided the experimental counterpart of the numerical findings, confirming the existence of three time-scales water dynamics, and the emergence of a layer of exceptionally slow water molecules around the lysozyme, in the presence of trehalose [15].

Here we expand upon these works to grasp new aspects of the hydration properties of ternary solutions where the amount of trehalose is effective for bioprotection by exploring: (i) the temperature dependence of spatial extension of the short-range perturbation induced by the presence of trehalose on a water/lysozyme solution, and (ii) the inter-molecular features in the THz region that are responsible for the collective vibrational modes of the constituents of the mixture. EDLS data are further complemented by Fourier Transform Infrared measurements in Attenuated Total Reflection configuration (ATR-FTIR spectroscopy). The ATR-FTIR profile was analyzed to isolate the solute-correlated (SC) spectrum [27], highlighting the effect of the sugar on the structure of the protein and the H-bonding properties of its hydration shell.

## 2. Materials and Methods

### 2.1. Samples

Lysozyme from chicken egg white lyophilized powder (M_w_ = 14.3 kDa) and D-(+)-trehalose dihydrate (M_w_ = 378.33) were purchased from Sigma-Aldrich and used without further purification. Binary (water-trehalose, WT) and (water-lysozyme, WL) and ternary (water-trehalose-lysozyme, WTL) solutions were prepared by weight, using doubly distilled and deionized water filtered in our laboratory. For the binary WT and WL solutions mole fractions of trehalose (x_T_) and lysozyme (x_L_), corresponding to x_T_ = 0.04 (40% in weight of trehalose) and x_L_ = 6.5 × 10^−5^ (4% in weight of lysozyme), respectively, were considered. The ternary WTL solution was prepared at x_L_ = 7 × 10^−5^ (3% in weight of lysozyme and 40% trehalose). Each solution was kept at 40 °C for 90 min under moderate stirring to ensure complete dissolving of the protein and/or sugar. The mixtures were then thermalized at room temperature and filtered through 0.2 μm filters before use.

### 2.2. EDLS

The EDLS spectra were collected over a frequency range from fractions to several thousands of GHz. The horizontally polarized scattered light is analyzed using two different spectrometers to explore such an extended spectral range. From 0.6 to 90 GHz, the low-frequency region was acquired by using a Sandercock-type (3 + 3)-pass tandem Fabry–Perot interferometer, characterized by a finesse of about 100 and a contrast higher than 10^10^. A 200 mW single mode solid-state laser at λ = 532 nm was used. Three different mirror separations, corresponding to different free spectral ranges, were used to obtain the depolarized spectra over the required frequency range [28]. The backscattering geometry was adopted to avoid intense contributions to the spectra coming from transverse acoustic modes [24,28]. The high-frequency region, from 60 to 36,000 GHz (2–1200 cm^−1^), was measured using a Jobin–Yvon U1000 double monochromator with 1 m focal length and equipped with holographic gratings. A 300 mW Ar+ laser operating on a single mode of the λ = 514.5 nm line was employed as the source. The detection system was a thermoelectrically cooled Hamamatsu model 943XX photomultiplier. The scattered light was analyzed by a 90° scattering geometry in two different frequency regions: from −10 to 40 cm^−1^ with a resolution of 0.5 cm^−1^ and from 3 to 1200 cm^−1^ with a resolution of 1 cm^−1^. More details are given in References [24,25,29]. After subtraction of the background contribution, low and high-frequency spectral signals were joined together, taking advantage of overlap of about half a decade in frequency. EDLS spectra are generally displayed by adopting susceptibility formalism. In order to do that, the imaginary part of the dynamic susceptibility χ″(ω) was calculated from the intensity of the depolarized scattered light, I_HV_(ω), through the relation χ″(ω) = I_HV_(ω)/[n_B_(ω) + 1], were n_B_(ω) is the Bose–Einstein occupation factor [23].

### 2.3. FTIR

Fourier transform infrared (FTIR) measurements in ATR configuration (ATR-FTIR) were performed with an Alpha (Bruker Optics) spectrometer. It was equipped with a GLOBAR source, a ROCKSOLIDTM interferometer, a KBr beam-splitter, an RT-DLATGS detector, and a high refractive index crystal (diamond). The Opus 7.5 Bruker Optics software was employed for spectral acquisition and analysis. The spectra were recorded in the 300–5000 cm^−1^ region by averaging 30 scans acquisition, with a resolution of 2 cm^−1^. The resulting spectra (ATR-absorbance) were corrected using the so-called extended ATR correction routine of the Opus software that accounts for the refractive index and wavelength dependences of the penetration depth [30], obtaining spectral distributions similar to those derived by measurements in transmission mode. The transmission-like spectra were baseline-corrected by simple subtraction of a constant offset. Based on the methods first developed by Ben-Amotz and coworkers [31], solute-correlated infrared (SC-IR) spectra were extracted by a direct spectral subtraction procedure, computing the difference between the spectrum of lysozyme solutions and the rescaled spectrum of the corresponding solvent. The rescaling factor is determined to give a final spectral distribution without negative components and with the minimum area [32].

## 3. Results

### 3.1. EDLS Data Treatment

EDLS measurements for both WTL and WT solutions were performed from 3.5 °C to 35 °C (namely, T = 3.5, 10, 14.5, 25, 35 °C). The susceptibility spectra, obtained after reconstruction of the entire profiles, are reported in Figure 1 as a function of frequency (ν = ω/2π). The spectra were normalized to the high-frequency peaks (>10 THz) corresponding to the Raman active modes of trehalose as visible in Appendix A.

Since the EDLS profiles cover more than four decades in frequency, many dynamical processes probed through fluctuations in the anisotropic components of the total polarizability of the system were detected in a single experiment. These processes involve both solute and solvent particle motions: from low to high frequencies, diffusional rotation of solute molecules, relaxation processes due to the restructuring of the water H-bond network, and intermolecular vibrational processes show up in the spectrum [26].

Over the last few years, a thoroughly worked out strategy has been developed to carefully analyze the data [15,26,33]. According to our experience, the susceptibility, χ″(ω), is modeled with a phenomenological function able to reproduce the whole spectral profile. For aqueous solutions, it is in general composed of three parts: $\chi^{\prime \prime }\left(\omega \right)=\chi_{SR}^{\prime \prime }\left(\omega \right)+\chi_{WR}^{\prime \prime }\left(\omega \right)+\chi_{VIB}^{\prime \prime }\left(\omega \right)$. The first term, $\chi_{SR}^{\prime \prime }\left(\omega \right)$, frequently reproduced by a Debye (D) function, is related to the rotational diffusion of the solute [26,29,34], the second term, $\chi_{WR}^{\prime \prime }\left(\omega \right)$, describes the water relaxations and is given by a sum of two Cole–Davidson (CD) functions [15,35], with a shape parameter fixed to 0.6 [15,33,35], the same value obtained in pure water (see Appendix A). Finally, in the third term, $\chi_{VIB}^{\prime \prime }\left(\omega \right),$ arises from the vibrational part of the EDLS spectrum and must be adapted on a case-by-case basis [15,22,26,29,34,35].

In the case of sugar-water mixtures, this contribution is mainly due to the H-bond intermolecular bending (1.5 THz) and stretching (5.1 THz) Raman modes of water and is usually modeled with two damped harmonic oscillator (DHO) functions [36,37]. Other terms besides these two DHOs have instead been used for more complex systems to include vibrational modes of the backbone and side groups. For lysozyme solutions, three Brownian oscillators (BO) are found to be able to effectively model the low-frequency protein vibrational modes [15,35,38].

Therefore, the full-spectrum data analysis of the EDLS spectra of WT and WTL mixtures was carried out following these guidelines. The global best-fit curves with their individual components are shown in Figure 2, denoting the very good agreement between model function and experimental data.

For the WT binary mixture, three spectral features of different origins characterize the experimental EDLS signal (violet line in Figure 2a). The low-frequency region of the spectrum is dominated by a strong long tail, which is attributed to light scattered by the rotational diffusion of single trehalose molecules (green dotted line) [39,40]. The central region, i.e., 10–100 GHz range, mainly arises from a dipole-induced-dipole scattering mechanism [33,40,41]. Here, the dynamics were related to local translations (H-bond restructuring dynamics) of bulk and hydration water molecules [24,40] (blue and cyan, colored areas). The motions of hydration water were slower than those of bulk water, being characterized by a retardation factor ξ~6, weakly temperature-dependent. We recall similar relaxation features in water lysozyme (WL) solutions, with ξ ranging from 6 to 8 depending on temperature and concentration [35,42]. For the sugar aqueous solutions, the two peaks at higher frequencies (lilac dashed line Figure 2a) were mainly assigned to H-bond bending (B) and stretching (S) intermolecular vibrations of water molecules [43].

The analysis of the WTL ternary mixture was, on the other hand, very complex. Over the whole range investigated, up to 10 THz, the acquired spectrum (dark blue line in Figure 1b) is more intense than that corresponding to the binary system. This was due to the lysozyme contribution that must be carefully considered. A method to obtain a very good reproduction of all the spectral signatures (orange line in Figure 2b) was developed by some of us [15] by combining the fitting results of WT binary data with that of the so-called difference spectrum (DS), as discussed in detail in Reference [15]. In particular, the residual component, obtained as a difference of WTL and WT spectra, after proper normalization, could bring out the protein contribution (see Appendix A), highlighting the presence of (i) the protein structural relaxation, well reproduced by a power law (PL, Figure 2b), ∝ ν^−0.3^ [35,44,45] and (ii) the solvent-free lysozyme vibrational modes ($\chi_{VIB}^{\prime \prime }{\left(\omega \right)}_{DS}$, Figure 2b), modeled with three BO functions, due to librational motions of solvent-exposed side chains and to backbone torsions [35,38,46], and also including the so-called “Boson peak,” arising from collective protein vibrations [38]. Beyond that, an additional component (see Appendix A) associated with a dynamics slowed down by a factor ξ~15–20 with respect to bulk water was detected, revealing the presence of a fraction of ultraslow water molecules in the trehalose-containing system [15]. It has to be stressed that this ultraslow component was not present in the spectrum of the binary WL solution, where only the protein hydration contribution with ξ~6–8 appears [35,42].

On the whole, the water relaxation dynamics in the WTL mixture was found to be characterized by three time scales, as depicted in Figure 2a: the smallest one related to the relaxation of bulk water (revealed in the WL and WT solutions and pure water), the intermediate one referring to the hydration water detected in binary solutions (WT and WL), and the longest scale associated to the ultraslow water in the ternary mixture. Despite the complexity of the data analysis, this three water-timescales picture was found to be supported by molecular dynamics (MD) simulations [16,21] that, through a selective calculation of density correlators, i.e., self-intermediate scattering function (SISF) of hydration water oxygen atoms), was capable of singling out the contribution coming from hydration water alone.

In addition to the retardation factors, we already analyzed the temperature dependence of these relaxation processes [15], highlighting that the apparent activation energy derived for each of them was about 8–10 kJ/mol. This suggested that in the temperature range investigated, these processes mainly originated from the local H-bonding dynamics of water. Activation energy values of a similar order of magnitude have also been obtained for H-bonding water rearrangement by other techniques probing density fluctuations, such as Brillouin light scattering (13 kJ/mol) and inelastic x-ray scattering (15 kJ/mol) [47,48].

New pieces of information regarding the perturbation induced by trehalose on the water molecules around lysozyme can be further gained by observing the temperature behavior of the hydration number *N*_ultraslow_, i.e., the number of water molecules per molecule of lysozyme that are dynamically involved in the ultraslow process. *N*_ultraslow_ can be obtained by evaluating the ratio between the intensity of the ultraslow component and that the three water components in the ternary mixture (considering the light scattering cross-section of all water molecules being approximately the same). That is, *N*_ultraslow_ = *r*Δ_ultraslow_/Δ_tot_, where *r* is the ratio of water to lysozyme molecules, Δ_ultraslow_ is the amplitude of the relaxation of *ultraslow* water, and Δ_tot_ = Δ_ultraslow_ + Δ_hydr_ + Δ_bulk_ is the total amplitude of water relaxation processes [15]. A simple reproduction of the water relaxation profile is shown in Figure 3a in comparison with the experimental counterpart $\chi_{WR}^{\prime \prime }\left(\omega \right)$ at 25 °C, showing a good agreement. The $\chi_{WR}^{\prime \prime }\left(\omega \right)$ profile is obtained by subtracting from the total spectrum $\chi^{\prime \prime }\left(\omega \right)$ the curves corresponding to the other trehalose and lysozyme contributions, as obtained by the fitting procedure (see Figure 2b). Calculation of *N*_ultraslow_ for the WTL mixture (Figure 3b) provides an average value of 710 ± 60; such an amount of water molecules closely matches the number of water molecules expected within a single layer around lysozyme [22,35,42]. As shown in Figure 3b, our results agree remarkably well with the values obtained from MD simulations for the same system [21]. These aspects of similarity between EDLS and MD results are impressive, considering the high number of degrees of freedom of the system under study and the fact that the EDLS technique probes the collective dynamics, while simulations (15,21) derive information from SISF. To have a more in-depth quantitative comparison, it will be extremely interesting to extract the collective dynamic structure factor from the time correlation function originated by simulations, and to play with parallels studying different ternary mixtures.

Therefore, in line with the numerical study, it is reasonable to assume the presence of a layer of very slow water entrapped between lysozyme and trehalose. Water molecules in this layer are much slower than in the bulk and slower than those in proximity to lysozyme in a binary WL solution when trehalose is not present in the mixture. Overall, even if it is not excluded, within a dynamic solvation process, some trehalose molecules may approach the lysozyme surface [21]. The existence of a subensemble of ultraslow water molecules under the influence of protein and sugar can reasonably be inferred by these results.

### 3.2. Terahertz Vibrations

More information can be acquired, on this point, by analyzing to what extent the presence of trehalose influences the low-frequency vibrations of lysozyme. This frequency/time scale, related to both solvent-solute H-bond rearrangements and amino acid torsions and librations, is very peculiar since it has been found to be connected with the onset of biological activity of hydrated proteins [38,49].

To single out the spectral feature related only to vibrations, $\chi_{VIB}^{\prime \prime }\left(\omega \right)$, the results of the full spectral analysis (Figure 2b) can be exploited. For better viewing, Figure 4a shows in double linear scale the total $\chi^{\prime \prime }\left(\omega \right)$ profile together with the tails of the curves reproducing the relaxation processes $\chi_{SR}^{\prime \prime }\left(\omega \right)+\chi_{WR}^{\prime \prime }\left(\omega \right)$ (colored areas). As a first step, the latter contributions were subtracted to the former, thereby isolating the vibrational contribution of the WTL spectrum (olive curve). The same operation was made on the WT and DS spectra (green and pink lines, respectively) to directly visualize, at each temperature, water-trehalose and lysozyme low-frequency vibrational modes. By construction, the total vibrational spectrum was given by the sum of these two contributions in the ternary system. We can notice that the whole temperature dependence comes from the WT contribution. In particular, Figure 4b shows that temperature keeps the B mode of water approximately unchanged while mainly affecting the S mode in a manner that resembles pure water [43,50]. Specifically, the intensity of the S-band was strictly connected to modifications on the local arrangement of H-bonded water molecules. This signal selectively detects the intermolecular vibration of a tetrahedral unit of five water molecules belonging to the H-bond network, and the width of the mode is related to the inhomogeneous distribution of H-bond interactions [43]. On this ground, the intensity reduction of the S-band could be attributed to the decrease of tetrahedral water units with temperature, consistent with literature data on water and water-sugar solutions [51]. As for the DS-vibrational contribution, we observed that its shape remained the same as the temperature changed. Furthermore, by comparing this vibrational band with the solvent-subtracted low-frequency Raman profile $\chi_{VIB}^{\prime \prime }{\left(\omega \right)}_{LYS}$ of the WL mixture at x_L_ = 6.5 × 10^−5^ and 25, 35, and 50 °C (cyan curves), we found that all the spectra generated a single master-curve meaning that the vibrational band of lysozyme in the ternary system was coincident with that in the binary mixture. Therefore, it was little or not at all perturbed by trehalose. In brief: in the THz frequency region, the additivity of $\chi_{VIB}^{\prime \prime }{\left(\omega \right)}_{WT}$ and $\chi_{VIB}^{\prime \prime }{\left(\omega \right)}_{LYS}$ was found to hold consistently with a preferential exclusion mechanism of cosolute from the protein hydration shell. Nevertheless, no signatures of structural changes of the ultraslow water subensemble were identified in this frequency range. In this respect, the fast intermolecular motion of the entrapped water molecules appeared unperturbed with respect to the bulk.

### 3.3. SC-IR Analysis

It was also interesting to analyze to what extent the structural properties of both protein and water, as probed by intramolecular modes in the 1000–4000 cm^−1^ range, were affected by the presence of trehalose. Ben-Amotz and coworkers introduced a methodology combining multivariate curve resolution and Raman spectroscopy that allowed one to extract the solute-correlated (SC) spectrum, which contains spectral features from the solute and the water molecules perturbed by the solute itself [27,31,52]. The method has been then extended to the analysis of FTIR spectra [53]. Here, solute-correlated infrared (SC-IR) spectra were extracted by a direct spectral subtraction procedure [31]. This was done by subtracting from the spectrum of lysozyme solutions the rescaled spectrum of the corresponding solvent to obtain the minimum-area non-negative spectral distribution [32]. The procedure was applied to both binary and ternary solutions, obtaining information on both the protein and its hydration water under trehalose’s influence in the solvation medium.

#### 3.3.1. Binary Mixture

Figure 5a shows the ATR-FTIR spectrum of the WL solution compared with the rescaled spectrum of neat water and the resulting SC-IR spectrum. This latter evidences the protein’s amide I and amide II bands at 1650 and 1550 cm^−1^, respectively, which are sensitive to its secondary structure [54,55,56]. In parallel, Figure 5b displays a comparison in the high-frequency region (2800–3800 cm^−1^)—sensitive to H-bonding interactions—between the spectrum of pure water due to the OH stretching modes SC-IR spectrum of the lysozyme solution, after normalization to the maximum intensity. Most of the SC-IR spectrum intensity accounts for the OH stretching vibrations of water molecules whose H-bonding features are affected by the solute. Contributions from the perturbed water are expected to be dominant in this OH stretching region compared to the direct contributions arising from the protein, mostly related to NH stretching vibrations [57]. This is qualitatively supported by comparing the SC-IR spectrum with the IR spectrum of solid lysozyme, normalized to the feature at around 2900 cm^−1^, due to the protein CH stretching modes. The comparison also underlines that the lysozyme signals would mainly affect the SC-IR spectrum at around 3200 cm^−1^. As a result, the intensity redistribution towards lower frequencies observed going from the spectrum of pure water to the SC-IR one could be explained considering the formation of relatively stronger H-bonds within the protein hydration shell than pure water. In other words, the protein seemed to induce an average strengthening of the H-bonds of hydrating water, as reflected by the reduction of the OH population involved in weaker H-bonds, present in neat water and resonating at higher frequencies (>3400 cm^−1^).

#### 3.3.2. Ternary Mixture

Figure 6a shows the spectrum of the ternary WTL solution, the rescaled spectrum of the binary WT solution, which represents the protein solvation medium in the ternary solution, and the resulting SC spectrum. In this case, this latter highlights the lysozyme amide bands at 1650 and 1550 cm^−1^, while a good compensation is achieved for the signals of trehalose, which are visible in the 1000–1500 cm^−1^ region of the solution spectra. The OH stretching region of the SC-IR spectrum is better visualized in Figure 6b; for the sake of comparison, the maximum-normalized spectrum of the neat water and the binary WT solution are also reported. The SC-IR spectrum of lysozyme in the ternary mixture is distributed to lower frequencies with respect to that of the solvent, represented by the binary WT solution. Thus, as before, the presence of lysozyme induces an average strengthening of H-bonding interaction, which would mainly relate to the formation of new OH···L links within the protein solvation shell. The presence of direct lysozyme contributions at around 3200 cm^−1^ is not expected to invalidate this qualitative conclusion.

Interestingly, the SC spectrum of lysozyme in the ternary WTL mixture compares reasonably well with the SC-IR spectrum of lysozyme in the binary WL system (Figure 7a), indicating the presence of trehalose does not cause any evident modification of the protein hydration features. Thus, the distribution of H-bonding interactions felt by lysozyme was not significantly altered by the presence of trehalose. This was also confirmed by looking at the amide signals (Figure 7b), which showed similar features in both SC-IR spectra. The absence of any clear signatures of specific trehalose/lysozyme interactions agrees with THz Raman data and with the idea that the protein remains strongly hydrated even in the ternary mixture (preferential hydration), in agreement with the water entrapment scenario [14]. Notice that, once normalized to the amide signals, the overall intensity of the band in the 3100–3600 cm^−1^ region (inset of Figure 7b) decreases significantly, going from the WL to the WTL solution. This suggested that the amount of water affected by lysozyme decreased from the binary WL to the ternary WTL solution, in qualitative agreement with EDLS results.

## 4. Conclusions

The joint use of depolarized light scattering and infrared techniques allowed us to add new information on the mechanisms by which water and trehalose molecules are accommodated around lysozyme in a WTL ternary mixture, at a trehalose concentration particularly effective in preserving biomolecules. A very broad frequency range was investigated to monitor several solute/cosolute and solvent processes at different time scales. To obtain a comprehensive view, WT and WL binary mixtures were analyzed compared to the ternary solution. The results presented here perfectly supplement our previous work [15]. By analyzing EDLS data over the relaxation dynamics time-scale (2–200 GHz in frequency), peculiar differences between binary and ternary solutions were revealed. Contrary to the WL solution, where an extended hydration shell of water molecules slowed down by a factor of 6–8 compared to bulk water is present around lysozyme, in the WTL mixture, EDLS revealed the presence of an extra water shell surrounding the protein of very slow molecules (15–20 times slower than that in the bulk water). Such a shell contains about 700 water molecules. Here, we show that this number, in agreement with molecular dynamics results [21], remains approximately constant upon temperature variation (from 35 °C down to 3.5 °C), thereby creating a sort of stable stress-protection coverage. This would support the preferential hydration of lysozyme with a negligible number of trehalose molecules close to the protein surface, which possibly favors the conditions for trehalose-mediated bio-preservation.

The analysis of the THz Raman region showed that the vibrational features of lysozyme are scarcely dependent on temperature and practically coincident in both ternary and binary mixtures. That is, the presence of trehalose does not influence the THz intramolecular modes of the protein.

To complement the depolarized light scattering data, we examined solute-correlated infrared (SC-IR) spectra of the binary (WL) and ternary (WTL) solutions to probe trehalose-induced perturbations on both water and lysozyme structures. In particular, the OH stretching and amide spectral regions sensitive to the H-bond organization of water and protein structure, respectively, were analyzed. SC-IR spectra proved that the protein causes an average strengthening of H-bonds of the hydration water, which maintains the same features also in the ternary mixture. Furthermore, the presence of trehalose does not affect the amide signals. Overall, our experimental findings did not evidence any clear signature of specific trehalose/lysozyme interactions. Such a result agrees with the view that the protein remains strongly hydrated even in the ternary mixture (preferential hydration) [58], providing support to the water entrapment scenario.

Noticeably, while the H-bonding rearrangement of this entrapped water is significantly slowed down at the picosecond time scale, its H-bonding structuring, probed by intermolecular modes (at THz frequencies) and intramolecular vibrations, is not influenced. As future prospects, the broad-band approach presented here can interestingly be extended to other protein/cosolute systems to investigate biopreservation and protein stability issues further. To this end, EDLS investigations could be effectively coupled with dielectric measurements (DS). DS, probing the reorientation of permanent molecular dipoles, can detect the rotational dynamics of both solute and solvent [59,60]. It will represent a powerful complementary tool to EDLS for studying hydration properties in extremely viscous media.

## Figures and Tables

**Figure 1 life-11-00995-f001:**
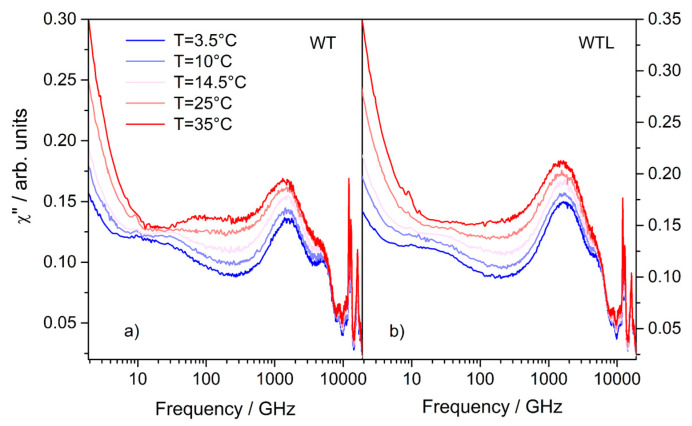
EDLS susceptibility spectra for WT binary (**a**) and WTL ternary (**b**) mixtures at the indicated temperatures.

**Figure 2 life-11-00995-f002:**
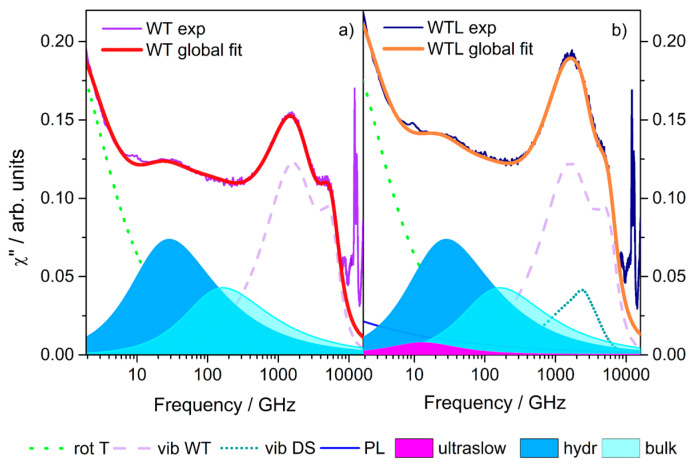
EDLS susceptibility spectra for WT binary (**a**) and WTL ternary (**b**) mixtures at 14.5 °C. Experimental and global best-fit curves with individual components are shown: rot T, the rotation diffusion of trehalose, PL the power-law describing the lysozyme relaxation, the sum of these two components giving the term $\chi_{SR}^{\prime \prime }{\left(\omega \right)}_{};$
vib WT, the vibrational contributions of the WT mixture, given by the sum of inter-molecular B&S modes, referred to as $\chi_{VIB}^{\prime \prime }{\left(\omega \right)}_{WT}$; vib DS, the vibrational contribution (arising from lysozyme) of the difference spectrum, namely $\chi_{VIB}^{\prime \prime }{\left(\omega \right)}_{DS}$, and finally superslow, hydration, and bulk water relaxations, the sum of which provides the term $\chi_{WR}^{\prime \prime }{\left(\omega \right)}_{}$
.

**Figure 3 life-11-00995-f003:**
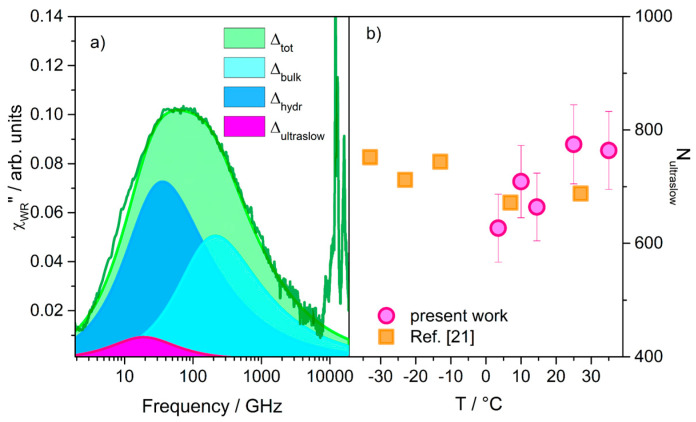
(**a**) $\chi_{WR}^{\prime \prime }{\left(\omega \right)}_{}$ difference profile (see text) and curves from the best-fit analysis that reproduce the water relaxation dynamics evidencing ultraslow, hydration, and bulk features. (**b**) The number of ultraslow water molecules hydrating lysozyme (circles) calculated as described in the text, and corresponding MD values (squares) estimated from Figure 2 of Reference [21].

**Figure 4 life-11-00995-f004:**
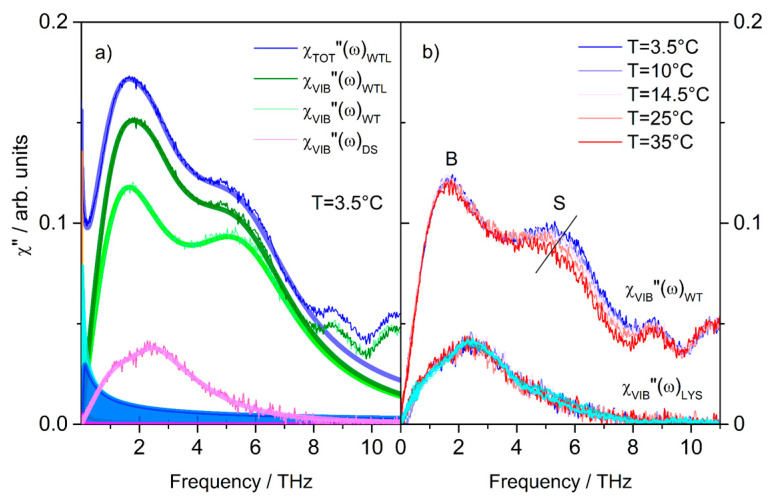
Inter-molecular vibrational modes of ternary and binary mixtures. (**a**) Total WTL spectra; Relaxations modes (colored areas); Relaxations-subtracted WTL, WT, and DS spectra ($\chi_{VIB}^{\prime \prime }{\left(\omega \right)}_{WTL}$
, $\chi_{VIB}^{\prime \prime }{\left(\omega \right)}_{WT}$, and $\chi_{VIB}^{\prime \prime }{\left(\omega \right)}_{DS}$, respectively) (**b**) Temperature dependence of relaxations-subtracted $\chi_{VIB}^{\prime \prime }{\left(\omega \right)}_{WT}$, and $\chi_{VIB}^{\prime \prime }{\left(\omega \right)}_{DS}$. Cyan curves correspond to $\chi_{VIB}^{\prime \prime }{\left(\omega \right)}_{LYS}$ spectra at 25 °C, 35 °C, and 50 °C, the three curves being indistinguishable, and practically coinciding with the $\chi_{VIB}^{\prime \prime }{\left(\omega \right)}_{DS}$ profiles at several temperatures.

**Figure 5 life-11-00995-f005:**
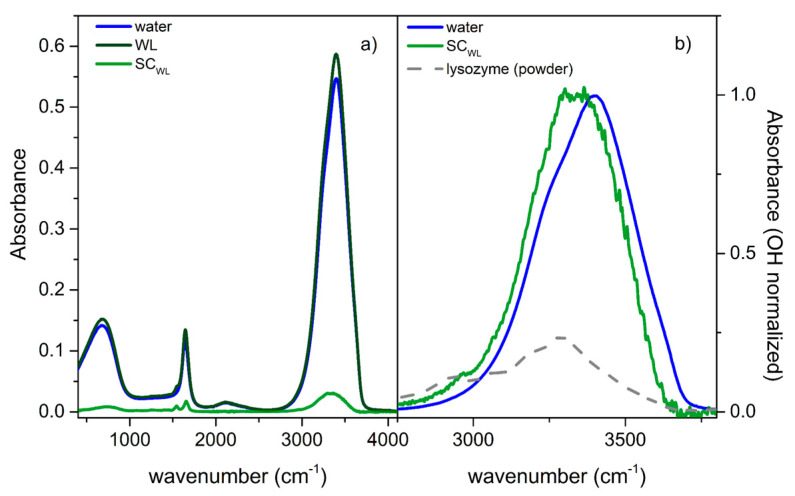
(**a**) ATR-FTIR spectrum of the WL solution (dark green line), the rescaled ATR-FTIR spectrum of neat water (blue line), and the resulting SC-IR spectrum (green line) (see the experimental section for details). (**b**) Comparison, in the OH stretching region, between the SC-IR spectrum and the maximum-normalized spectrum of neat water (blue line). The ATR-FTIR spectrum of solid lysozyme, normalized at around 2900 cm^−1^ (CH stretching signal), is also reported.

**Figure 6 life-11-00995-f006:**
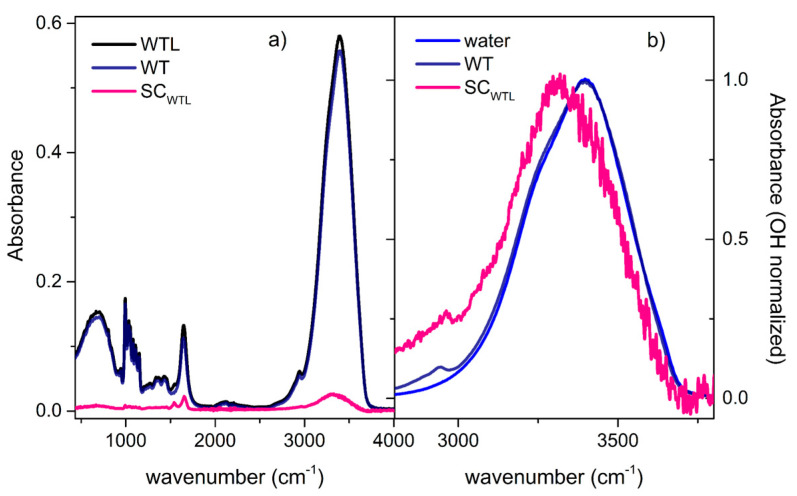
(**a**) ATR-FTIR spectrum of the ternary WTL mixture (black line), the rescaled ATR-FTIR spectrum of the binary WT solution (dark blue line), and the resulting solute-correlated (SC-IR) spectrum (magenta line). (**b**) Comparison, in the OH stretching region, among the SC-IR spectrum (magenta) and the maximum-normalized ATR-FTIR spectra of the binary trehalose/water solution (black line) and of neat water (blue line).

**Figure 7 life-11-00995-f007:**
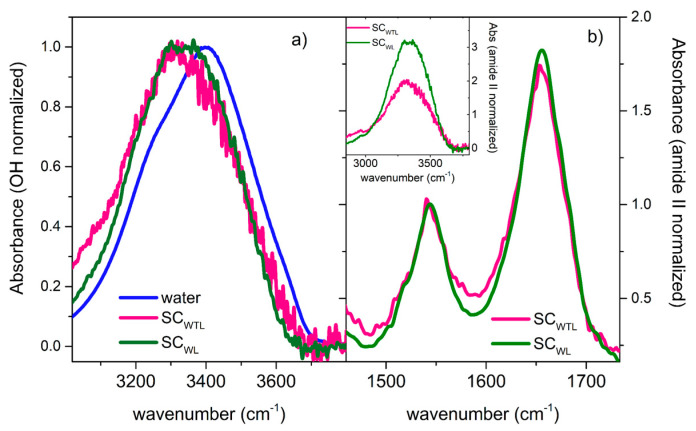
(**a**) Comparison among the SC-IR spectra obtained for the ternary WTL (magenta line) and binary WL (green line) solutions, together with the ATR-FTIR spectra of neat water (blue line). The spectra are normalized at the maximum intensity. (**b**) SC-IR spectra of the ternary WTL (magenta line) and binary WL (green line) solutions in the spectral region of amide I and amide II bands. The spectra are normalized on the total area of the two bands. The corresponding spectra in the 3000–3800 cm^−1^ region are depicted in the inset.

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
