# Peer review of "Protein Hydration in a Bioprotecting Mixture"

_life, 2021, doi:10.3390/life11100995_

Round 1
Reviewer 1 Report
The work appears significant and provides important evidence for the interactions between protein and the compatible osmolyte, trehalose. However, the experimental evidence is not adequately explained and insufficiently justified for the interpretation leaving open the possibility for alternative explanations. The authors chose to leave out the Water-Lysozyme EDLS spectra, incluting only water-Trehalose and the ternary mixtures, yet they indicate that the WL spectrum also includes ultraslow water, this undermines the primary conclusion that trehalose is impacting the enrapped water within the ternary system. In fact, little justification of the changes in hydration water is presented for the EDLS data. I feel strongly that figures 1 and 2 be updated to include the WL spectrum for comparison so that the differences in ternary solution can be identified.
The manuscript is also in need of significant grammatical editing. The authors exhibit excessive use of commas and run on sentences that impact the readability of the document. The reviewer has included many suggestions in the attached pdf, but doubts that all of the errors were caught.
Below the reviewer has identified five concerns that I would like to addressed prior to publication.
Specific Concerns:
1) As the analysis is presented, I am not convinced the ultraslow component is necessary to fit the data. It seems that a slight shift in the hydration data, or in the dynamics of the bulk water in the ternary solution could also fit the data exceptionally well. The addition of the third component of water dynamics is not well justified in the analysis explanation.
2) It appears that the number of ultraslow waters is independent of temperature from -30 to +30 degrees. This seems very perplexing, and in fact would imply that there are no dynamic changes to this technique over this temperature range? Would that not imply that this is a static artifact of the spectrum and not a feature of the water, which one would expect to change substantively in this range?
3) While I am not surprised that trehalose did not impact the structure of Lysozyme, I am surprised that no changes in the Lysozyme vibration were observed in changing from 25 to 50 degrees. It seems that side chains and more loosely structured loops and turns would experience an increase in motions that would impact the vibrational spectra. Could the author comment?
4) Some form of concentration dependence or by comparing with other proteins of compatible solutes would really validate this work. As presented with only a single concentration and exhibiting no temperature dependence, it is difficult to demonstrate that this additional signal is caused by ultraslow water molecules, and not an experimental artifact. Would it be possible to complete an additional such as 15 wt% and show an increase in the number of ultraslow water with concentration?
5) Another possible interpretation is that these techniques are insensitive to the trehalose-protein interactions because the signals are dominated by other contributions. The authors have not adequately eliminated this possibility.

Author Response
We thank the Referee for carefully reading the manuscript (ms) and for her/his valuable comments which we have addressed in the attached response. The Referee’s comments are given in italics, the Authors’ reply as normal (red) text. The changes made are marked up using the “Track Changes” function in the revised ms. Moreover, to make some parts of the ms clearer, an additional file with supporting information (SI) has been prepared, including new figures (namely Figure S1, S2, and S3).

Reviewer 2 Report
Fascinating research on the hydration of the lysozyme molecule in the presence of trehalose. Trehalose induces significant changes in the structure of the water, forming the shell on the surface of the protein.
The question is whether the change in system viscosity has been taken into account. For example, the addition of trehalose may increase the viscosity of the system and thus the mobility of the molecules in the tested system. Moreover, it seems interesting if the CD spectroscopy would not show changes in the secondary structure of the protein under the influence of sugar addition and thus confirm the stability of the structure.
This is only a comment that does not change the high grade of the submitted work.
Author Response
We thank the Referee for carefully reading the manuscript (ms) and for her/his valuable comments which we have addressed in the attached response. The Referee’s comments are given in italics, the Authors’ reply in normal (red) text. The changes made are marked up using the “Track Changes” function in the revised ms. Moreover, to make some parts of the ms clearer, an additional file with supporting information (SI) has been prepared, including new figures (namely Figure S1, S2, and S3).

Reviewer 3 Report
I am pleased to be made aware of this work. The authors are using depolarised light scattering in the GHz region and IR spectroscopy to investigate the ternary mixtures of lysozyme, water and, trehalose in the context of biopreservation.
The analyses of IR difference spectra of binary and ternary mixture suggest that there is no structural change due to lysozyme and trehalose interactions but a general increase in h-bonding strength.
The authors claim that in addition to slow water molecules surrounding the lysozyme additional even slower molecules appear when trehalose is added. They interpret that as a strengthening of the water hydration shell of the lysozyme and an advantage for biopreservation.
This work is a continuation of the very interesting previous light scattering work of the authors. They complement published data with further IR spectroscopic information. The analyses and conclusions pose some questions to me, which I think could lead to some useful revision of the manuscript.
Questions:
Which geometry (vertical-horizontal) was used for the depolarised light scattering? Which laser power is used? (Is stated in the literature, only to be sure for this publication.)
How precisely is the susceptibility in fig. 1 normalized?
What is the CD shape parameter of the fitted curves in general and fig. 2? Why are they chosen? How would the resulting analyses change if chosen otherwise?
How is in fig. 2b) accounted for lysozyme rotation? It is only stated that in fig. 2a) the slow contribution (green) is due to trehalose reorientation.
Why is the relaxation between 10 and 100GHz pure water relaxation and not a beta relaxation of trehalose? It is a very flexible molecule with a strong scattering signal. Did you measure pure trehalose?
Line 224: Is there activation energy with a quote that can be compared with the 8-10kJ/mol?
Line 334: "Most of the intensity of the SC-IR spectrum accounts for the OH stretching 333 vibrations of water molecules whose H-bonding features are affected by the solute." Why is a water contribution in the SC-IR spectra? The SC-IR was meant to remove it? Maybe rephrasing?
Suggestions:
A log-log representation of fig. 2 would enable us to get an idea of existing powerlaws.
Maybe one could be more explicit with the very useful comparisons to MD. Would be more instructive what one can learn and whatnot.
I miss an additional source for the basics of depolarised light scattering like J. Berne and R. Pecora: "Dynamic Light Scattering".
Main challenge:
The signal of lysozyme and trehalose is magnitudes stronger than the signal of depolarised light scattering in water (As well proven by a cited work of the authors). Therefore depolarised light scattering is a technique to rather not see water dynamic in general. The argument why the 10-100GHz peak is attributed exclusively to water, even when there should be contributions by the other molecules as well, is not enough made. I see arguments in the cited literature but they should be made more explicit and discussed in the manuscript.
Water has a strong dipole moment and shows up in dielectric experiments very strongly. The combination of these techniques can distinguish water from other contributions as demonstrated in: "Dynamics of aqueous peptide solutions in folded and disordered states examined by dynamic light scattering and dielectric spectroscopy" PCCP 2021.
Do you see hinds for nanobubbles? Maybe a description of how clear the samples are would be nice. Maybe for further technical aspects, the authors might look into: "Depolarized Dynamic Light Scattering and Dielectric Spectroscopy: Two Perspectives on Molecular Reorientation in Supercooled Liquids".
I very much like the experiments and strongly recommend rethinking the analyses in light of my commands and suggested literature. After revision, I strongly recommend reconsidering the manuscript. I very much looking forward to your reply and send my very best regards!
Author Response
We thank the Referee for carefully reading the manuscript (hereinafter referred to as ms) and for her/his valuable comments which we have addressed below. Hereafter, the Referee’s comments are given in italics, the Authors’ reply as normal (red) text. The changes made are marked up using the “Track Changes” function in the revised ms. Moreover, an additional file with supporting information (SI) has been prepared, including new figures (namely Figure S1, S2, and S3).

Round 2
Reviewer 1 Report
The new version of the paper is significantly improved in terms of the presentation of the data analysis. I think the supplementary figures provide insight that is much needed to support the primary paper figures. The grammar and flow of the paper has also improved significantly.
Reviewer 3 Report
Thank you for the information provided in your response and paper. I recommend the manuscript for publication.